# Three-Dimensional Bioprinting of Decellularized Extracellular Matrix-Based Bioinks for Tissue Engineering

**DOI:** 10.3390/molecules27113442

**Published:** 2022-05-26

**Authors:** Chun-Yang Zhang, Chao-Ping Fu, Xiong-Ya Li, Xiao-Chang Lu, Long-Ge Hu, Ranjith Kumar Kankala, Shi-Bin Wang, Ai-Zheng Chen

**Affiliations:** 1Institute of Biomaterials and Tissue Engineering, Huaqiao University, Xiamen 361021, China; 19013081049@stu.hqu.edu.cn (C.-Y.Z.); 19013081060@stu.hqu.edu.cn (X.-Y.L.); 20014087023@stu.hqu.edu.cn (X.-C.L.); 1814122010@stu.hqu.edu.cn (L.-G.H.); ranjithkankala@hqu.edu.cn (R.K.K.); sbwang@stu.hqu.edu.cn (S.-B.W.); 2Fujian Provincial Key Laboratory of Biochemical Technology, Huaqiao University, Xiamen 361021, China

**Keywords:** 3D bioprinting, bioink, decellularized extracellular matrix, tissue engineering

## Abstract

Three-dimensional (3D) bioprinting is one of the most promising additive manufacturing technologies for fabricating various biomimetic architectures of tissues and organs. In this context, the bioink, a critical element for biofabrication, is a mixture of biomaterials and living cells used in 3D printing to create cell-laden structures. Recently, decellularized extracellular matrix (dECM)-based bioinks derived from natural tissues have garnered enormous attention from researchers due to their unique and complex biochemical properties. This review initially presents the details of the natural ECM and its role in cell growth and metabolism. Further, we briefly emphasize the commonly used decellularization treatment procedures and subsequent evaluations for the quality control of the dECM. In addition, we summarize some of the common bioink preparation strategies, the 3D bioprinting approaches, and the applicability of 3D-printed dECM bioinks to tissue engineering. Finally, we present some of the challenges in this field and the prospects for future development.

## 1. Introduction

Tissue engineering, a cutting-edge field of science, utilizes cells, scaffold materials, and growth factors to construct biologically active tissues in vitro for organ replacements using various biomanufacturing techniques, including electrostatic spinning [1,2,3], microfluidics [4,5,6], and 3D bioprinting [7,8,9], among others [10,11]. However, the fabrication of scaffold materials as an extracellular matrix (ECM) has been a significant impediment to their transition to clinics [12]. Furthermore, achieving a high degree of ECM mimicry by using either synthetic materials (poly (ethylene glycol), Pluronic F127, etc.) or natural materials and their derivatives (collagen, hyaluronic acid, gelatin, methacrylate gelatin, etc.) remains a significant challenge [13,14]. The recent advancements in 3D bioprinting decellularized extracellular matrix (dECM)-based bioinks and the relevant decellularization strategies are reviewed in this article.

The ECM is a complex network of macromolecules, which provides a site for cell survival and activity, as well as the ability to regulate cell behavior [15]. In addition, the ECM, to some extent, mimics the cellular microenvironment and provides a three-dimensional space for cells [16,17,18]. As a result, in vitro reconstruction of the ECM is crucial for engineering tissues. Owing to these aspects, decellularization technology has garnered enormous interest in regard to fabricating natural ECM. By removing the cellular components from tissues and organs while preserving the composition, biological activity, and integrity of the ECM, this approach dramatically enriches the content of tissue engineering scaffold materials [19]. The resultant material after decellularization, in combination with the previously mentioned polymer-based materials, can be used as a tissue engineering scaffold material.

Compared with traditional tissue engineering methods, the emerging 3D bioprinting technology has the advantages of controlled design of structures and high material utilization, and offers unique advantages in the personalized processing of biomaterials. With the advent of 3D bioprinting technology, it is now possible to create cell-laden 3D structures with different geometries for personalized tissue repair and organ fabrication [20,21]. A bioink, which is essentially a biological material used to wrap cells in 3D printing, primarily mimics the ECM [22]. In recent years, 3D bioprinting of dECM-based bioinks has emerged as a hot research topic, with many novel bioinks [23,24,25] and novel manufacturing methods [26,27]. This review critically emphasizes various aspects of the ECM, decellularization methods, bioink preparation strategies, 3D bioprinting methods, and tissue engineering applications of dECM-based bioinks. The overview diagram of the article is shown in Figure 1.

## 2. Extracellular Matrix (ECM)

### 2.1. Components

The ECM provides a suitable site for cell survival and activity, while it also influences cell shape, metabolism, function, migration, proliferation, and differentiation through signal transduction systems [15,16]. The ECM is made up of an intricate network of different macromolecules, which can be broadly classified into four major groups: collagen, glycoproteins (non-collagenous), glycans (aminoglycan and proteoglycan), and elastins (Figure 2) [28]. The ECM is found in lower concentrations in epithelial tissues, muscle tissues, and the brain and spinal cord, while in higher concentrations in the connective tissues [29]. The components of the ECM and their assembly are often determined by the cells from which they arise, and are tailored to the specific functional needs of the tissue. The ECM of the cornea, for example, is a clear, soft lamella, whereas tendons are tough as a rope. The ECM not only provides support, attachment, water retention, and protection to the cells, but it also has a wide range of dynamic effects on it [16].

Fibrous proteins (collagen and elastin): Collagen is the most abundant protein in animals, accounting for nearly 30% or more of total body protein [30]. It is a framework structure in the ECM that can be synthesized and secreted extracellularly by fibroblasts, chondrocytes, osteoblasts, and specific epithelial cells. Collagen is found throughout the body in various organs and tissues. Elastin is a critical protein found in the ECM, primarily acting to keep the tissues and organs physiologically functioning as they stretch and flex [31]. Elastin is composed of two types of short peptides alternately arranged: a short hydrophobic peptide that gives the molecule its elasticity, and an alpha helix with alanine- and lysine-rich residues that form cross-links between adjacent molecules. Elasticity is the most important physicochemical property of elastin.

Adhesion proteins (fibronectin and laminin): Fibronectin (FN) is a large glycoprotein found in all vertebrates, with a molecular sugar content ranging from 4.5% to 9.5% and a glycan chain structure that varies depending on one tissue cell origin and differentiation status [32]. In the ECM and on the cell surface, FN exists in an insoluble form, and intermolecular cross-linking via disulfide bonds allows the attachment of cells to the ECM and the subsequent assembly into fibers. FN, unlike collagen, does not spontaneously form fibers; instead, it is guided by cell surface receptors and is only found on the surface of specific cells (e.g., fibroblasts) [33]. Laminin (LN) is a large glycoprotein that, like the basement membrane with type IV collagen, forms the basement membrane. Notably, it is the earliest component of the ECM to appear in embryonic development. Meanwhile, LN is a glycoprotein with a high sugar content (15–28%), with approximately 50 N-linked glycoconjugates, and is the most complex glycoprotein with the most complex glycoconjugate structure known to date [34]. Moreover, the multiple receptors of LN are recognized and bound to its glycoconjugate structure.

Glycoproteins (glycosaminoglycan and proteoglycan): Glycosaminoglycan (GAG) is a polysaccharide with an unbranched long chain made up of repeated disaccharide units. Based on the constituent glycosyl groups, the connection method, the degree of sulfation, and the location, amino glycans are classified into six types: hyaluronic acid (HA), chondroitin sulfate, dermatan sulfate, acetyl heparin sulfate, heparin, and keratan sulfate [35]. Except for HA and heparin, several other amino glycans are covalently bound to core proteins to form proteoglycans. These proteoglycans are covalently linked to amino glycans (other than HA) with core protein. Several polymorphs of proteoglycan can have molecular weights of 108 KD or higher and can exceed the size of bacteria [36]. For example, aggrecan, a cartilage component, contains a GAG composed primarily of chondroitin sulfate (CS) and keratan sulfate (KS).

Matrix receptors: Integrins are the most common cell surface receptors that mediate cell adhesion to the ECM. Integrins are made up of two chains, α and β, in which the α chain comprises 1420 amino acids, while the β chain consists of 840 amino acids [37]. Integrins are an important class of ECM protein receptors that, on one hand, can bind to the ECM or other cell surface ligands and mediate cell–cell and cell–ECM interactions; while on the other hand, can bind to cytoskeletal proteins or intracellular signaling molecules through their intracellular regions. In summary, integrins are involved in cellular messaging, cell cycle regulation, cell shape, and cell motility, in addition to their mechanical effects across membranes [38,39].

### 2.2. Biological Roles

The ECM not only serves physical functionalities, such as connectivity, support, water retention, stress resistance, and protection, but also assists in a variety of biological functions in the basic life activities of cells [16].

The ECM influences cell survival, growth, and death: Apart from mature blood cells, most normal eukaryotic cells must adhere to a specific ECM to inhibit apoptosis and survive [40]. Notably, the epithelial and endothelial cells detached from the ECM often undergo programmed death [41]. Cell proliferation is affected differently by altered extracellular matrices. For instance, fibroblasts, on one hand, proliferate faster on fibronectin substrates and slower on laminin substrates compared with epithelial cells, which respond to fibronectin and laminin proliferation in contrary ways [42]. Tumor cells, on the other hand, lose their reliance on fixation dependence and proliferate in a semi-suspended state.

Shape determination: The shape of the cell is determined by the extracellular matrix to which it adheres. The same cell types could take on completely different shapes when adhering to different extracellular matrices. To this end, epithelial cells adhere to the basement membrane to demonstrate their polarity. The role of the ECM in determining cell shape is accomplished through its receptors influencing the cytoskeleton assembly [43]. Different cells with different extracellular matrices mediate different cytoskeletal assemblies, resulting in various shapes.

Control cell differentiation: Often, cells differentiate by interacting with specific ECM components [44]. For example, myogenic cells proliferate and remain undifferentiated in fibronectin; whereas in the presence of laminin, they stop proliferating, differentiate, and fuse into myotubes [45].

The ECM contributes to cell migration: The ECM regulates cell migration speed and direction, serving as a scaffold for cell migration [46]. For example, fibronectin promotes fibroblasts and corneal epithelial cell migration, while laminin promotes the migration of many tumor cells [47]. Moreover, chemotaxis and chemotactic migration rely on the ECM, implying embryonic development and wound healing. Notably, cell adhesion and cytoskeleton assembly are required for cell migration, in which the cell adhering to a specific ECM causes the formation of adhesion patches, which are the rivets connecting the ECM to the cytoskeleton.

The ECM influences all life phenomena, such as cell shape, structure, function, survival, proliferation, differentiation, and migration. Therefore, it is important in all physiological activities, including morphogenesis and organ formation during embryonic development, and in maintaining the structural and functional perfection of the adult body (including immune response and trauma repair) [16].

## 3. Decellularization Methods and Evaluation

### 3.1. Decellularization Methods

The goal of decellularization is to remove all the cellular components from a tissue or organ while preserving the composition and integrity of the natural ECM [19]. In this context, various factors, such as cell type, tissue density, thickness, and lipid content, determine the effectiveness of a tissue decellularization method. Broadly speaking, various decellularization methods are classified into physical, chemical, or biological approaches based on the type of processing and the application of precursor materials. Although classified into different types, a combination of physical, chemical, and biological enzymes is frequently used to improve the efficiency of decellularization. In this section, we present details of all these decellularization strategies, highlighting the factors affecting their decellularization behaviors, along with their pros and cons in comparison with other methods.

#### 3.1.1. Physical Methods

Physical decellularization works on the basic principle of mechanically disrupting the cell membrane structure of cells in tissues. The changes in the cell membrane structure cause undesirable biochemical reactions, and continued treatment results in cell death and the subsequent decellularization of tissues via solution washing, nucleic acid, and lipid removal [48]. Although physical methods alone have been successful in removing cells from a small percentage of tissues, they are often used in conjunction with chemical and biological methods (described in later subsections) to remove genetic material residues from scaffolds more effectively. Several common physical methods include the freeze–thaw method, the mechanical stirring method, and supercritical fluid (SCF) extraction [48].

Freeze–thaw: Rapid freezing often results in cytoplasmic crystals forming intracellularly, disrupting the cell membrane and causing cell lysis. Notably, this procedure usually requires multiple cycles of freezing and thawing to achieve better results. Although physical methods can effectively preserve the ultrastructure of the ECM, temperature control can significantly affect the integrity of the ECM. In addition, cyclic freeze–thawing alone does not completely elute the cellular components, and further processing in combination with chemical or biological methods is required [49,50].

Mechanical stirring: Mechanical stirring is one of the most commonly used decellularization methods. Typically, the desired tissue for decellularization is immersed in chemical reagents, decontaminants, or enzymes, and then subjected to mechanical agitation, thereby destroying the cell structure to release cellular material for decellularization purposes [51]. In fact, the choice of reagents, the order of use, the concentration of reagents, and the time, speed, and strength of agitation need to be adjusted according to the characteristics of the tissue or organ of different origin. However, it is worth noting that the application of the mechanical stirring method requires reasonable control of the stirring conditions to achieve adequate protection of the structural integrity and mechanical properties of the extracellular matrix.

Supercritical CO_2_-based extraction: When the phase state of gases changes above a critical temperature and pressure point, it transforms into a new type of fluid known as a supercritical fluid. The most widely used supercritical fluid is supercritical carbon dioxide, whose safe, optimal critical conditions and eco-friendly, non-toxic processing are easy to achieve. Several reports utilizing this innovative technology have demonstrated the feasibility of decellularization in fabricating dECM [52,53,54].

#### 3.1.2. Chemical Methods

Chemical-based decellularization, the most commonly used method, is achieved by dissolving cell membranes and degrading DNA using chemical reagents (acids, bases, surfactants). Some of the main chemical treatment methods and reagents are described as follows.

Acids and bases: In general, the acidic solutions separate DNA from ECM by dissolving cytoplasmic components and degrading nucleic acids. In addition, acids denature ECM proteins, including GAGs, collagens, and growth factors. Some common examples of acid reagents used for decellularization include acetic acid, peroxyacetic acid (PAA), acetic acid, hydrochloric acid, sulfuric acid, and deoxycholic acid [55,56]. To this end, alkaline solutions denature chromosomes and plasmid DNA. Moreover, alkaline solutions can disrupt the cross-linking of collagen fibers and weaken the mechanical properties of decellularized ECM. Various commonly used alkaline reagents include sodium hydroxide, ammonium hydroxide, sodium sulfide, and calcium hydroxide [57,58].

A descaling agent is another commonly used chemical decellularization method. It can be classified as ionic, non-ionic, or amphoteric, with the latter primarily introducing ionic and non-ionic types. Various descaling agents can effectively dissolve cell membranes and decellularize by destroying proteins in the ECM, which inevitably influences the ultrastructure of the ECM [59]. Ionic detergents act effectively on cell membranes, cytoplasm, and the nucleus, dissolving cell membranes, lipids, and DNA, and disrupting protein–protein linkages. Among various descaling agents, sodium dodecyl sulfate (SDS) is the most commonly reported ionic detergent for chemical-based decellularization [60]. To this end, the non-ionic descaling agents disrupt inter-lipid and inter-lipid–protein linkages while preserving protein–protein linkage integrity, making them more suitable for thinner tissues. Although Triton X-100 [61] is a typical example of this class, utilizing a non-ionic descaling agent is often ineffective during decellularization.

#### 3.1.3. Enzymatic Methods

Biological enzymes selectively cleave cell adhesion proteins, and separate and lyse cells from the surrounding matrix. Unfortunately, prolonged enzyme treatment degrades matrix components, such as collagen, elastin, and glycosaminoglycans. Moreover, the residual enzymes in the dECM may result in potential adverse reactions [58]. Therefore, the complete elution of various chemical and biological reagents is essential after the decellularization process. We summarized the decellularization approaches of various tissues and organs, as shown in Table 1.

### 3.2. Evaluating the Prepared dECM

The decellularized matrix is derived from homologous or allogeneic tissues or organs, and it may contain residual cellular components that cause immune rejection. It is critical to establish or standardize strict criteria for evaluating the prepared decellularized matrix for safety reasons. Accordingly, the decellularized extracellular matrix is often evaluated primarily in terms of nucleic acid analysis [97], cytoplasmic or non-nucleic component analysis [98], protein analysis [99], and mechanical or structural analysis [100].

Based on the available studies on decellularized matrices applied for in vivo or in vitro studies, a uniform standard was established to evaluate the residual nucleic acid content to avoid immune rejection. The standards were stated as less than 50 ng dsDNA/mg dry weight of dECM, less than 200 base pair DNA fragment length [101], and minimal or no nucleic acid material observed by histological or immunohistochemical analysis. Furthermore, spectroscopy-based methods, antibody-based component-specific staining, and ELISA can be used to identify essential components such as collagen, GAGs, and adhesion proteins [102]. Similarly, the surface morphology of the dECM must be observed. Decellularized samples are typically vacuum-dried, and gold sprayed before being examined using a scanning electron microscope [103].

## 4. Strategies for Preparing dECM-Based Bioinks

Bioinks are often referred to as biomaterial-wrapped cells in direct cell printing, providing appropriate support and a three-dimensional microenvironment for cells. According to the source and affinity, primary bioink materials can be divided into natural exo-matrix materials of animal origin (collagen, fibrin, hyaluronic acid, GelMA, matrix gum, etc.), natural biomaterials of non-animal origin (alginate, chitosan, agarose, etc.), and synthetic polymer materials (polyethylene glycol, Pluronic F127, etc.). Decellularized matrices are usually compounded with the above materials to enable the fabrication of three-dimensional structures. In this context, several neoteric methods of constructing decellularized substrates are briefly described (Figure 3). Visscher et al. [23] constructed a photo-cross-linkable cartilage-derived ECM bioink for auricular cartilage tissue engineering. Briefly, the prepared decellularized cartilage tissue powder was digested in an acetic acid solution of porcine pepsin. To make a photo-cross-linked dECM-based bioink, the separated product was initially dissolved in acetic acid, the pH was then adjusted to 8–9, and methacrylic anhydride (MMA) was added. After the reaction, the photo-cross-linkable decellularized matrix was obtained after dialysis and lyophilization (Figure 3A). Zhuang and colleagues [24] prepared a composite bioink consisting of GelMA, dECM, and nano-clay that possessed better printability and biocompatibility than dECM-based bioinks. In general, conventional methods use pepsin to digest the decellularized matrix before compounding the hydrogel material, resulting in the degradation of the natural structure, the biochemical components, and the mechanical strength of the decellularized tissue. In an attempt, Kim and group [25] developed a type of dECM powder–based bioink and successfully fabricated micro-patterns with 93% cell viability to overcome the relatively poor printability and mechanical properties of traditional dECM bioinks. In this research, liver dECM powder was prepared without pepsin treatment instead of freeze-milling processes, and then loaded into gelatin to manufacture a 3D structure as shown in Figure 3B. In addition, Zhao and colleagues investigated the effect of different digestion times on the properties of dECM. They discovered that dECM possessed a high viscosity at the initial stage of digestion (3 h), as well as good printability and tissue-induced regeneration ability [66].

## 5. 3D Bioprinting Technologies

3D bioprinting, a novel manufacturing technology, uses cells and biological materials as a bioink to establish hierarchical three-dimensional structures with complex structures and biological functions through additive manufacturing methods, according to the requirements of bionic morphology, organism function, and cellular microenvironments [104]. 3D bioprinting has been widely used in the past few decades to build many tissues and organs, such as skin [105], cartilage [106,107], and liver [81,108,109], not only for patients suffering from diseases, but also for drug screening [110,111,112], organ transplantation [113], and other research. Nevertheless, there are still many bottlenecks in 3D bioprinting: the development and application of bioinks [114,115], bioprinting of vascularized structures in vivo [116,117], and achieving functionalization of printed structures. Herein, we systematically describe several printing methods using dECM-based bioinks: extrusion-based [118], inkjet [119], and digital light processing [120]. In brief, extrusion bioprinting builds structures by extruding bioink to form continuous fibers; droplet bioprinting generates discrete droplets for stacking and molding; and light-cured bioprinting uses photosensitive materials for light curing and stacking layer-by-layer to generate 3D models.

Extrusion-based bioprinting is currently the most common method due to its ease of use, a wide range of material selection (polymer melt, hydrogel, dECM, nano-clay, etc.), and low application cost [121]. Typically, bioink is deposited onto a printing platform by pneumatic or mechanical assistance (piston or screw) in a syringe or particular cartridge. In addition, extrusion-based bioprinting can be adapted to create vascular structures with coaxial nozzles. Despite the advantages and successes, the drawbacks are relatively obvious, such as a low print resolution [122] and shear force affecting cell viability [123]. To solve the poor extrusion printability of dECM bioinks, the use of multiple material composite, especially nanoparticles, has become a widespread approach. Shin et al. [124] developed a dECM-based bioink mixed with Laponite and PEGDA to improve the viscosity of the system (above 5000 Pa·s). Laponite not only ensured smooth extrusion during the manufacturing process, but also maintained high fidelity during the stacking process. In another case, beta tri-calcium phosphate [125] and graphene oxide (GO) [126] were used to improve the printability of dECM-based bioink. Apparently, dECM compounded with alginate [127,128], GelMA [129,130], and gelatin [131,132], presented good extrudability with improve printability. A schematic diagram of several extrusion printing types of devices is shown in Figure 4A.

Compared with extrusion-based bioprinting, inkjet bioprinting is based on the micro-electro-mechanicalsystems (MEMS) process with thermal bubble or piezoelectric-driven jet micro drop molding. This approach offers the advantages of low cost, high accuracy, and fast molding speed [133]. Continuous inkjet and drop-on-demand (DOD) printing approaches are the two most common types of currently employed inkjet printing. Among them, DOD printing is further divided into thermal DOD inject bioprinting, piezoelectric DOD inject bioprinting, electrostatic DOD inject bioprinting, and electrohydrodynamic jetting. The two most common inkjet printer device diagrams are as shown in Figure 4B. However, the thermal effects and mechanical stresses generated by inkjet bioprinting technology during the printing process can damage the encapsulated cells and reduce cell survival. Furthermore, inkjet bioprinting technology cannot produce high-viscosity materials, making it incapable of printing high cell density hydrogels, limiting the development of inkjet bioprinting [134].

Digital light printing (DLP) is an upgraded version of SLA (stereo lithography appearance). DLP works by projecting product cross-sectional graphics onto the surface of liquid photosensitive resin using digital micromirror elements to project, allowing the irradiated resin to be light-cured layer-by-layer, resulting in a relatively fast printing speed (Figure 4C) [135]. Furthermore, by non-utilizing conditions such as shear stress and higher temperatures and pressures, DLP technology is gentler on cells and bioactive components. In addition, diverse materials such as photo-cross-linkable resins, ceramics, and dECM can be brought into printing. Owing to the superiority of DLP bioprinting technology, several outstanding DLP printers have emerged in recent years, resulting in tremendous advancements in fabricating 3D structures.

## 6. Applications

Cells in the human body are hierarchically arranged in a complex and dynamic microenvironment, referred to as the ECM, in which various growth factors and other cells present different effects on cell behavior [136]. Therefore, establishing a biomimetic ECM is critical for developing tissue repair, artificial organs, and drug screening models. In this section, we present the applications of decellularized bioinks with a focus on the most recent reports.

### 6.1. Cartilage-Derived dECM Bioinks

Cartilage is a non-vascular, non-lymphatic tissue in the body, densely packed with connective tissues. However, it possesses minimal self-repair ability when subjected to external injury or long-term chronic strain [137]. Cartilage dECM bioinks, derived from specific native tissue, have been applied for use in cartilage tissue repair efficaciously.

In one instance, scaffolds were fabricated with a mixture of PU and PCL polymers and cell-laden decellularized meniscus ECM (me-dECM) bioink (Figure 5). Briefly, me-dECM bioink was first prepared after decellularization of the porcine medial meniscus and validated for relevant composition (Figure 5i) and rheological properties (Figure 5ii). Subsequently, the scaffold with a simulated meniscus structure was prepared by combining the use of a magnetic resonance (MRI) imaging technique to scan the meniscal articular bone, using PU_PCL material to simulate the meniscus structure (Figure 5iii), and wrapping the me-dECM bioink with human marrow mesenchymal stem cells (hBMSCs) in the printed meniscus. The results indicated that me-dECM bioink with high printability and long-term architectural integrity performed well in meniscus tissue recapitulation [138]. However, a further difficulty in developing a 3D tissue structure that mimics the microstructure and microenvironment of natural cartilage tissue is that various pro-inflammatory factors can impede tissue regeneration [133]. In an attempt to address this problem, another research group compounded cartilage dECM with poly(ethylene glycol) diacrylate (PEGDA) and combined it with the natural anti-inflammatory molecule honokiol (Hon) to create cartilage scaffolds using 3D printing technology. The levels of pro-inflammatory factors TNF-α, IL-1β, and IL-6 released from macrophages co-cultured with PEGDA/ECM scaffolds were significantly increased after lipopolysaccharide (LPS) treatment. However, the addition of Hon could significantly inhibit the secretion of the above pro-inflammatory factors, indicating that Hon had excellent anti-inflammatory effects. Moreover, in vitro animal experiments revealed that the PEGDA/ECM/Hon scaffold promoted the regeneration of cartilage and bone tissue at the site of osteochondral defects [63]. Apart from inflammatory issues, the cross-linking mode of hydrogels (ionic cross-linking, photo-cross-linking, enzymatic cross-linking) can also impact the restorative effect. Although UV cross-linking is the most commonly used modality for GelMA hydrogels, studies revealed that UV light potentially influenced cellular activity [139]. To address this issue, SF-dECM blends were mixed with the same volume of 80% PEG for in situ cross-linking to create a cross-linker-free bioink with similar biological and mechanical activities to the original cartilage. Finally, the SF-dECM bioink-fabricated 3D cartilage scaffold promoted BMSC proliferation and facilitated chondrogenesis [64].

### 6.2. Liver-Derived dECM Bioinks

Although the liver has a strong ability to regenerate itself, hepatocytes cultured in vitro rapidly lose their phenotypic characteristics and functions in vivo, which significantly limits the research on fabricating artificial livers and liver cancer designs for drug screening [79]. Therefore, a culture platform that mimics the in vivo environment of hepatocytes is urgently needed. In this context, many efforts to simulate the in vivo microenvironment of hepatocytes to address liver diseases have been explored. In 2011, Ren and colleagues used liver dECM as a three-dimensional culture substrate for hepatocytes and demonstrated that the dECM could promote cell proliferation while maintaining phenotype and function [140]. On the downside, this simple in vitro 3D culture had not allowed for the precise deposition of cells and the customization of personalized tissues. Mao et al. developed a fresh decellularized bioink composed of GelMA and liver dECM (Figure 6i), in which human hepatocyte cells were encapsulated to fabricate an inner gear-like structure of liver microtissue (Figure 6iii) using DLP-based 3D bioprinting (Figure 6ii). In vitro experiments revealed that the dECM played a prominent role in enhancing hiHep cell activity, proliferation, and liver function metabolism. Furthermore, DLP 3D bioprinting technology showed higher print resolution than extrusion printing [68]. Although UV cross-linking can improve print resolution and result in faster cross-linking, the effects on cells are complex and difficult to ascertain.

### 6.3. Skin-Derived dECM Bioinks

Similar to cartilage and liver dECM-based bioinks, skin dECM-based bioinks present the advantages of a tissue-specific microenvironment and tissue repairing. However, weak mechanical strength leads to low printability and high molding difficulty [141]. In 2018, Kim et al. prepared a skin decellularization matrix and carried out a detailed investigation of its gelation ability. It was observed that S-dECM bioink was in a pre-gel state at 15 °C and could be fully cross-linked after 30 min of incubation at 37 °C (Figure 7A(iii)). Next, an endothelial progenitor cell (EPC)-laden 3D-printed skin patch was fabricated to verify the capability to promote wound healing and vascularization [142]. Notably, the results of this study provided an essential reference for the preparation of skin tissue engineering bioinks. Won and coworkers [80] configured a bioink using skin dECM and human dermal fibroblasts to print artificial skin tissue structures, and cross-linked them through temperature changes. By analyzing the gene expression pattern in the cells of the construct, the skin regeneration mechanism of the bioink was verified, and the successful demonstration of the decellularized matrix was able to effectively enhance the skin morphology and the development-related gene expression (Figure 7B).

### 6.4. Cardiac-Derived dECM Bioinks

A 3D-printed pre-vascularized stem cell patch was reported to enhance the therapeutic efficacy of myocardial injury in 2016 [143]. Prior to this work, Jang and colleagues [144] developed a novel printing and cross-linking method for cardiac decellularized matrix bioinks to investigate the potential of dECM for cardiac repair. A two-step cross-linking method using sequential vitamin B2-induced UVA cross-linking and thermal gelation to solidify decellularized extracellular matrix (dECM) bioink was applied to print cardiac decellularized matrix bioinks in a bid to achieve a precise modulation of the mechanical properties of the printed structures. As shown in Figure 8A, the decellularized matrix bioink mixed with VB2 was extruded and then induced using UV irradiation to covalently cross-link the protein components in dECM to form a three-dimensional structure with specific mechanical properties, followed by further cross-linking at 37 °C. Accordingly, Jang et al. developed a stem cell patch for cardiac tissue regeneration. Briefly, decellularized bioinks encapsulating human c-kit + cardiac progenitor cells (hCPCs) and human turbinate tissue-derived mesenchymal stem cells (MSCs), respectively, were extruded onto pre-printed PCL substrates using a dual-jet printer to prepare cardiac patches, aimed at investigating whether they could compensate for the shortcomings of conventional stem cell therapy. As a result, both in vitro culture tests and animal experiments showed excellent therapeutic effects. In particular, the stem cell cardiac patch showed strong vascularization ability in in vivo trials and significantly improved heart function. Das et al. [145] prepared heart dECM-based bioinks encapsulating primary cardiomyocytes, and fabricated engineering heart tissue (EHT) models with a high elastic modulus using a dual-jet printer (Figure 8B). Specifically, the matrix microenvironment and culture conditions are decisive factors affecting cell–cell and cell–matrix interactions, affecting not only the structural arrangement of cardiomyocytes, but also the expression of related genes.

### 6.5. Blood Vessel–Derived dECM Bioinks

Gao [146] fabricated a bio-blood-vessel structure to deliver endothelial progenitor cells (EPCs) and the proangiogenic drug atorvastatin for the treatment of ischemic diseases (Figure 9). The EPCs and atorvastatin-loaded poly(lactic-co-glycolic) acid (PLGA) microspheres (APMs) were encapsulated by a hybrid bioink composed of vascular tissue–derived decellularized ECM (VdECM) and extruded using 3D coaxial cell printing technology. During the printing process, Pluronic F127/CaCl2 (CPF-127) components were extruded into the inner layer, and VdECM/alginate was placed as the outer layer. Finally, CPF 127 was removed after ionic cross-linking to form a hollow vessel structure. Further, an evaluation of the therapeutic effect in an in vivo model in nude mice revealed enhanced cell proliferation and differentiation of EPCs, increased neovascularization, and a significant salvage of ischemic limbs, indicating that 3D-printed ECM-mediated cell/drug implantation presented a new reference for the treatment of ischemic diseases.

### 6.6. Kidney-Derived dECM Bioinks

Ali et al. [147] imparted photo-cross-linking properties to kidney dECM grafted with methacrylic anhydride to print functional kidney microtissues in vitro without the support of other polymers as shown in Figure 10. Methacrylate-modifiable cartilage dECM bioinks have been reported in a previous study [23], in which the methacrylic anhydride content could regulate the mechanical strength of the printed structures by controlling the grafting rate of the dECM. Therefore, it is appropriate to predict that photo-cross-linked dECM bioink will be a popular direction for future research.

## 7. Possible Challenges and Solutions

It is undeniable that dECM materials play an important role as tissue engineering scaffold materials; however, there are still many challenges and problems that need to be overcome in the preparation of dECM and the preparation of dECM-based bioinks. We aim to summarize these problems to find solutions and optimization in further research work.

The main challenges of the dECM are toxicity, mechanical properties, and immunity. The dECM of allogeneic origin must undergo strict sterilization procedures to ensure the maximum avoidance of side effects caused by the material’s toxicity [148]. However, numerous studies have shown that inappropriate sterilization methods can lead to negative effects on the structure, degradation, and biological activity of the dECM, and can even result in the production of new toxic substances. For instance, gamma radiation causes damage to the structure, and mechanical properties [149] and ethylene oxide cause protein damage, and even carcinogenesis [150]. Therefore, optimizing the sterilization of the dECM still needs to be studied in-depth with regard to the appropriate method and time of sterilization for different tissues. In addition, the mechanical properties of the dECM have been a long-standing issue. This article summarized cases of decellularized tissues and organs from different sources. Indeed, most of them were digested with pepsin, leading to severe damage to the natural structural and mechanical properties of the dECM [25]. Therefore, more work is needed to balance the biological and mechanical properties of the dECM. As the predominant allogeneic or xenobiotic donor, the dECM may cause immune-related issues, which could be a great challenge for achieving long-term in vivo safety [151]. Although there has been consensus in the evaluation of the dECM, different tissues induce altered thresholds of the cellular content of the host immune-inflammatory response. Thus, it is necessary to test the remaining cellular components, such as mitochondria [152]. As the mechanisms of the relationship between specific cellular components and the host response become better understood, the criteria for evaluating the effect of decellularization should be updated and refined accordingly.

In addition, we summarized many other challenges to be overcome in the bioprinting of dECM bioinks, in terms of printability and vascularization regeneration capabilities. Although some methods have been proposed to enhance their printability, the weak mechanical properties and slow cross-linking speed of conventional dECM bioinks make them impossible to manufacture in high-precision micro and nanostructures or in gradient structures [25,153,154]. We believe that the development of photocurable double bond–modified [147] or thiol-modified dECM bioinks can solve this problem to a certain extent. Finally, the vascularizing regenerative capacity of the dECM is crucial in tissue repair and regeneration. Though most tissues and organs are structured with rich vascular networks, several studies indicated that the pro-vascularization of the dECM is not very promising [63]. Combining dECM bioinks with pro-angiogenic–related nanomaterials or drugs can effectively modulate the angiogenesis of recruited progenitor cells or embedded stem cells, and this phenomenon has also been demonstrated in other studies [155,156].

## 8. Conclusions and Future Perspectives

Currently, tremendous advancements have been evidenced in the field of generating scaffolds for tissue engineering and tumor models for drug screening based on the 3D printing of dECM-based bioinks. The functional characteristics of scaffolds printed with dECM bioinks from different organ tissues have been validated, and the morphology and properties of the printed scaffolds have been defined as the matured preparation parameters and printing parameters. In this review, we summarized the advances in the 3D bioprinting of dECM-based bioinks, including scaffolds, artificial tissues and organs, and tumor models, among others. Finally, we summarized the main challenges regarding the dECM and dECM bioinks that are currently being faced and proposed some solutions. In conclusion, the dECM is a highly promising tissue engineering material, and we sincerely hope to formulate more standardized decellularization evaluation criteria in the future, to develop dECM-based bioinks with controlled mechanical, degradation, and biological properties, to build tissues and organs using 3D bioprinting technology, and to create more success stories for the field of life medicine.

## Figures and Tables

**Figure 1 molecules-27-03442-f001:**
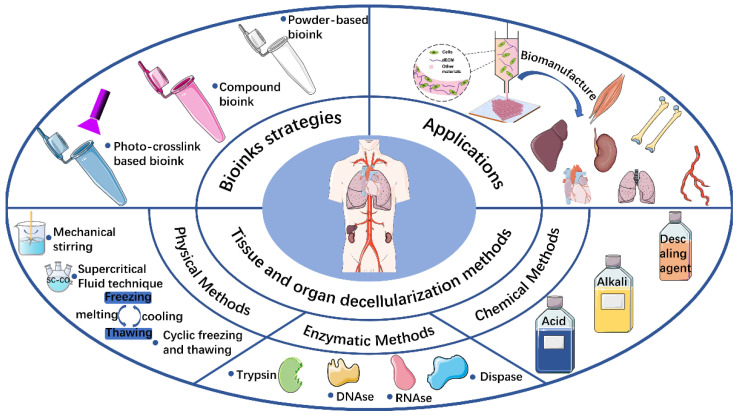
Schematic illustration highlighting the various decellularized extracellular matrix (dECM) preparation methods, and the bioink preparation strategies and applications of 3D bioprinting dECM-based bioinks in tissue engineering.

**Figure 2 molecules-27-03442-f002:**
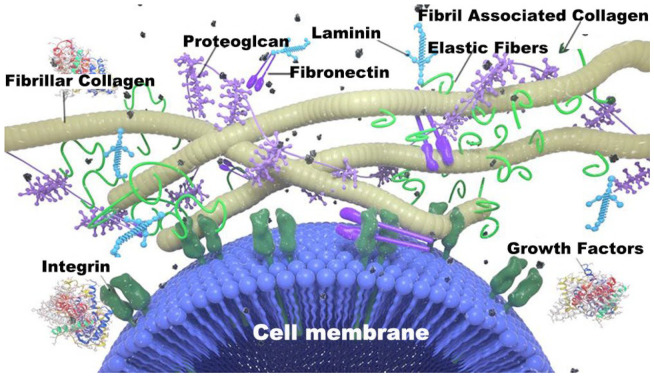
Representative illustration of extracellular matrix (ECM) compositional layout indicating cellular engagement with ECM biomolecules and primary components of general ECM space [28].

**Figure 3 molecules-27-03442-f003:**
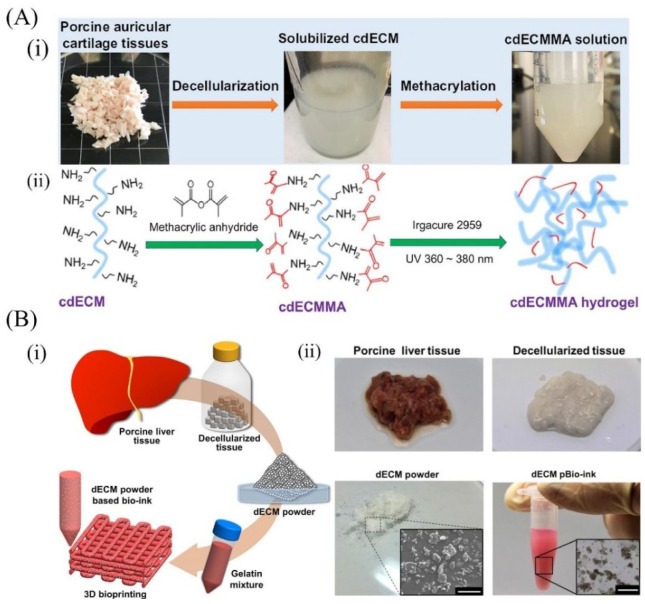
Strategies for preparing decellularized bioinks. (**A**) Illustration of cdECMMA bioink formulation containing cells and three-dimensional (3D) bioprinting process using cell-laden cdECMMA bioink: (i) Preparation of cdECMMA; and (ii) schematic diagram of preparation mechanism of cdECMMA [23]; (**B**) Schematic representation of the preparation of dECM powder–based bioink (i) and its application to bioprinting (ii) [25].

**Figure 4 molecules-27-03442-f004:**
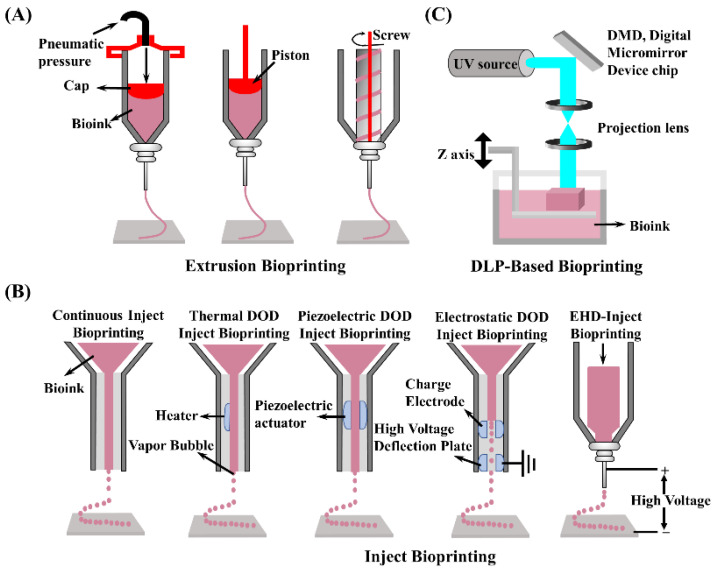
Schematic diagram of three representative 3D bioprinting technology devices: (**A**) extrusion bioprinting; (**B**) inject bioprinting; (**C**) DLP bioprinting.

**Figure 5 molecules-27-03442-f005:**
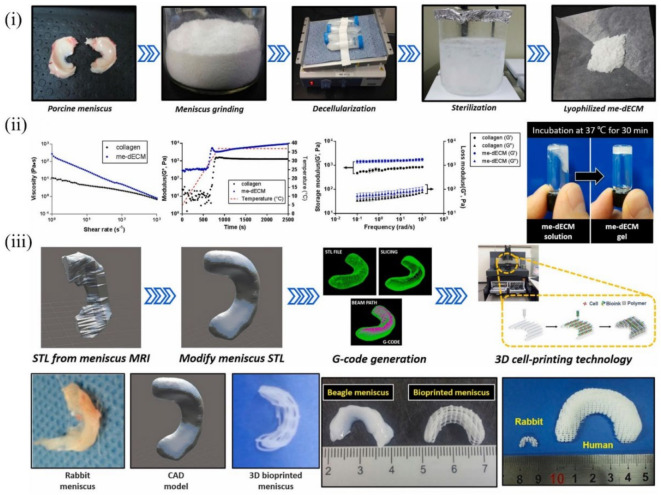
3D bioprinting of biocompatible and functional meniscus constructs using meniscus-derived bioink: (**i**) decellularization process of meniscus; (**ii**) rheological properties of me-dECM bioink and COL bioink; (**iii**) CAD-based 3D bioprinting of diverse meniscus constructs of rabbit, beagle, and human models [138].

**Figure 6 molecules-27-03442-f006:**
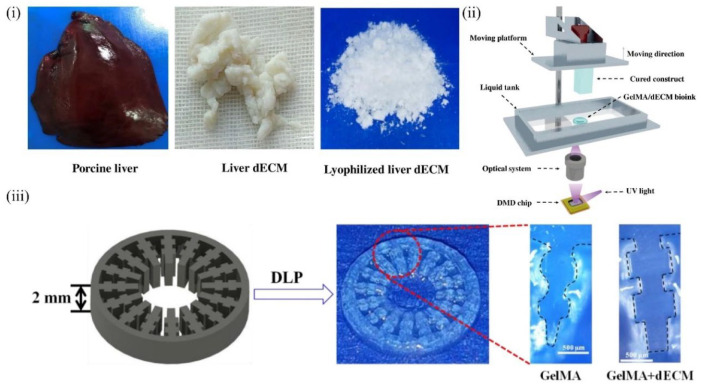
A typical example of liver-derived dECM bioink for 3D printing application: (**i**) digital image of fresh porcine liver/liver dECM/lyophilized liver dECM; (**ii**) schematic of DLP-based 3D printer; (**iii**) designed liver microtissue model and DLP printing results [68].

**Figure 7 molecules-27-03442-f007:**
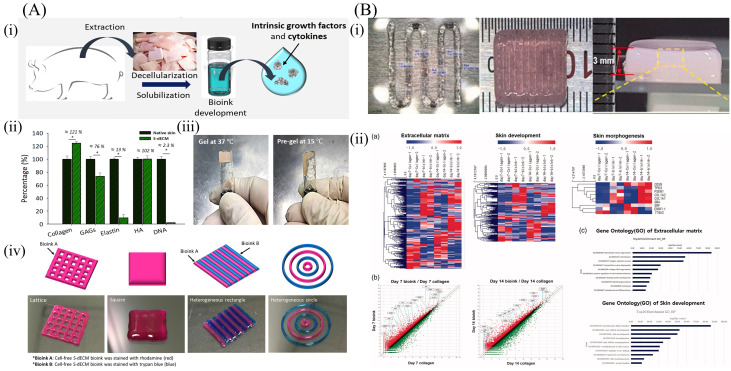
Two examples of skin-derived dECM bioinks for 3D printing applications. (**A**) Skin-derived bioink formulation and its properties analysis: (i) S-dECM bioink preparation process; (ii) quantitative analyses of dECM bioink, including collagen, GAGs, elastin, hyaluronic acid, and DNA; (iii) sol-gel transition of dECM bioink; and (iv) printability test of dECM bioink [142]; (**B**) Structure of the 3D-printed construct using skin bioink and gene expression: (i) cell-laden 3D scaffold; and (ii) changes in gene expression in the 3D-printed cell-laden construct [80].

**Figure 8 molecules-27-03442-f008:**
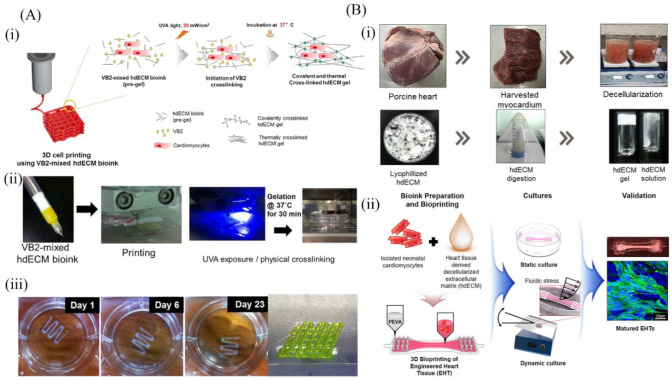
Two examples of cardiac-derived dECM bioinks for 3D printing applications. (**A**) Schematic illustration of a new cardiac-derived dECM bioink for 3D printing: (i) schematic illustration of a two-step cross-linking mechanism that applies concurrent cross-linking of vitamin B2-induced covalent cross-linking and thermal cross-linking; (ii) 3D printing and cross-linking; and (iii) digital image of the scaffold [144]; (**B**) Schematic depicting the stages starting with the preparation of the hdECM bioink to fabrication of the cell-laden EHT: (i) development of the hdECM bioink; and (ii) fabrication of the cardiomyocyte-laden EHT using a 3D bioprinter [145].

**Figure 9 molecules-27-03442-f009:**
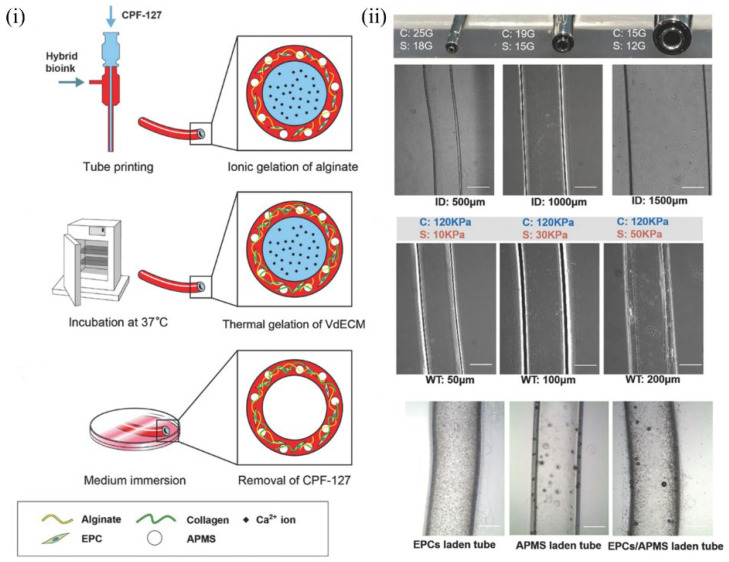
Schematic of research strategy: (**i**) schematic diagram of manufacturing coaxial vascular device and materials; (**ii**) schematic diagram of the coaxial blood vessel [146].

**Figure 10 molecules-27-03442-f010:**
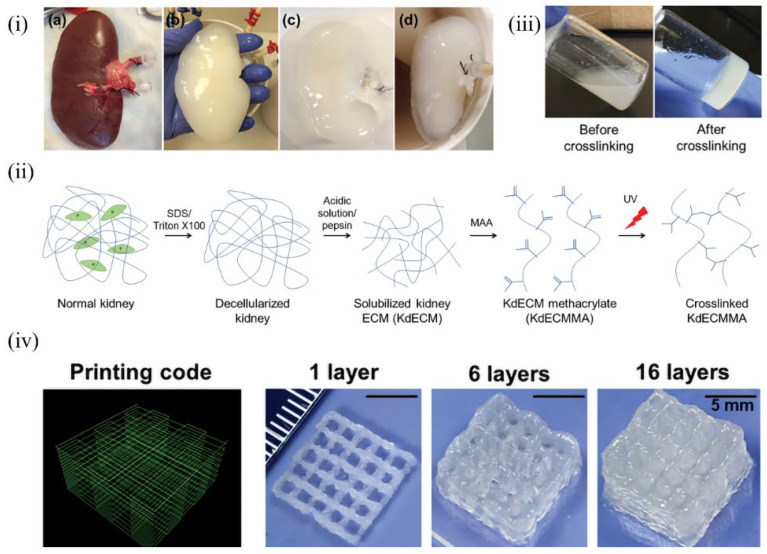
Preparation of KdECM and KdECMMA-based bioink formulations: changes in gene expression in the 3D-printed cell-laden construct: (**i**) gross images of the decellularization process: (a) normal kidney, (b) SDS treatment for 36 h, (c) Triton X-100 treatment for 24 h, and (d) washing in saline for 72 h; (**ii**) schematic illustration of a photo-cross-linkable kidney-specific ECM hydrogel; (**iii**) photography of KdECMMA before and after UV cross-linking; (**iv**) printing code and gross images of the printed KdECMMA-based constructs [147].

**Table 1 molecules-27-03442-t001:** Overview of decellularized methods.

Tissue or Organ Sources	Decellularized Method	Mode of Digestion	Ref.
Porcine lateral and medial menisci	Frozen for 5 min and thawed at 21 °C for 10 min 6 times; 0.25% trypsin for 8 h; 3% SDS for 72 h; 50 U/mL DNAse in PBS for 48 h	0.1% peracetic acid	[62]
Porcine cartilage tissue	Freeze–thaw cycles 3 times; 1% Triton X-100 for 1 d; immersed in 1% SDS for 24 h; 200 U/mL DNase I for 12 h	0.5 M acetic acid with 30 mg of pepsin for 48 h	[63]
Goat articular cartilage tissue	0.1% EDTA and 3.5% PMSF for 24 h; 1% Triton X-100 in Tris-HCl (pH = 7.5) with a protease inhibitor cocktail for 24 h; 50 U/mL DNAse and 1 U/mL RNAse for 12 h	1 mL of 0.1 M HCl containing 1 mg of pepsin for 48 h	[64]
Human auricular cartilage	4% SDS for 3 h; 1000 U/mL DNase for 3 h	-	[65]
Porcine tendon tissues	100% acetone for 30 min; 0.25% trypsin-EDTA; 2% SDS for 96 h	3 mg mL^−1^ pepsin in 0.1 M HCl	[66]
Porcine auricular cartilage	Immersed in 0.02% Tris/EDTA with protease inhibitor for 48 h; 1% Triton X-100; incubated with DNAse/RNAse (15 μg/mL) for 24 h; retreated with 0.02% Tris/EDTA solution for 48 h	-	[67]
Porcine liver	0.025% trypsin for 30 min 1% Triton solution for 24 h; 2% SDS for 36 h	Digested in 0.5 M acetic acid and pepsin solution for 96 h	[68]
Rat liver	1% Triton X-100 for 2 h; 0.1% SDS for 1 h; 750 U/mL DNAse and 25 U/mL RNAse for 30 min	Digested in 1 mg/mL of HCl (0.1 M) of pepsin for 72 h	[24,69]
Porcine liver	0.5% Triton X-100 for 9 h; 1% SDS for 3 h	-	[69]
Rat liver	1% Triton x-100 with 0.1% NH_4_OH (15 mL/min, 1 h; 20 mL/min, 2 h); sterile DI water (5 mL/min, 40 min; 15 mL/min, 15 min; 20 mL/min, 45 min); 0.1% peracetic acid (PAA) in 4% alcohol (5 mL/min, 40 min); submerged in PAA (30 min); sterile DI water (5 mL/min, overnight)	-	[70]
Porcine, canine, human, rat liver	Exposed the liver tissue to trypsin/EGTA and Triton X-100	Digested in pepsin solution	[71]
Porcine liver	0.1% SDS wash overnight	Digested at a 10 mg/mL dECM and 1 mg/mL pepsin at 0.01 M HCl for 48 h	[72]
Porcine skin	0.25% trypsin for 6 h; 70% ethanol for 10 h; 3% H_2_O_2_ for 15 min; 1% Triton X-100 in 0.26% EDTA/0.69% Tris for 6 h with a solution change for an additional 16 h; 0.1% peracetic acid/4% ethanol for 2 h	Digested in a 1 mg/mL pepsin solution in 0.01 N HCl for 48 h at 10 mg ECM/mL solution	[73]
Porcine skin	0.25% trypsin for 6 h; 1% Triton X-100 for 24 h; 10% isopropanol for 24 h; 30 U/mL DNase for 24 h; 0.1% peracetic acid in 4% ethanol for 2 h	Digested in papain solution (125 μg/mL) for 16 h	[74]
Nile tilapia skin	2.5 U/mL disperse for 3 h; 1% SDS for 6 h; 25 U/mL Pierce Universal Nuclease for 3 h; 1% SDS for 1 h	-	[75]
Groin skin	Cycle freeze–thaw 3 times; 0.25% trypsin/EDTA for 2 h; processed with isopropanol overnight; treated with 1% Triton X-100 for 48 h	-	[76]
Porcine peritoneum	Treated with a solution (pH 5.6) containing 2% SDS and 0.3% NaCl; ultrasonic treatment for 24 h;	-	[77]
Porcine small intestinal submucosa	Treated with mechanical removal of the tunica mucosa, the tunica serosa, and the tunica muscularis externa; treated with peracetic acid to remove remaining cells, RNA, and DNA	-	[78]
Porcine skin	0.25 wt% trypsin and 1 mM EDTA for 6 h; 1 wt% TritonX-100 for 24 h; 30 U/mL DNase for 24 h	0.5 M acetic acid solution containing 15 mg of pepsin per 100 mg dECM for 120 h	[79]
Porcine lateral and medial menisci	Frozen in liquid nitrogen for 5 min and then thawed at 21 °C for 10 min repeated 6 times; 0.25% (*w*/*v*) trypsin for 8 h; 3% (*w*/*v*) sodium deoxycholate for 3 d; 50 U/mL DNAse for 48 h	Lyophilized and pulverized into fine powder	[80]
Rat heart	Perfused through the ascending aorta with 200 mL of PBS containing heparin (20 U/mL) and 10 mM adenosine followed by 0.1% SDS, deionized water, 1% Triton X-100, 100 U/mL penicillin-G (Gibco), 100 U/mL streptomycin, and 100 U/mL amphotericin B	-	[81]
Porcine heart	0.1% SDS containing 7 mmol/L EDTA for 24 h, washed with 70% ethanol	2.0 mL of 6.0 N HCl for 24 h	[82]
porcine aortic valves and pericardia	5 mM Tris buffer with 1% Triton X-100 for 24 h; HBSS medium supplemented with 100 mg/L DNase, 20 mg/L RNase and 100 mg/L trypsin for 90 min; new 5 mM Tris buffer with 1% Triton X-100 for 24 h	-	[83]
Porcine myocardium	PBS solution with 1.0% Triton X-100 for 72 h; 20 mg/mL ribonuclease A and 0.2 mg/mL deoxyribonuclease for 48 h	0.05% collagenase, type IV, 0.5 mg/mL pancreatin, 1 mg/mL BSA solution	[84]
Zebrafish ventricular wall	Repeated freeze–thaw cycles, red blood cells, and DNA/RNA are removed by the erythrolysis buffer and deoxyribonuclease/ribonuclease	Mechanically ground into fine powders in liquid nitrogen	[85]
Porcine vena cava	0.1% SDS for 16 h; 40 U/mL DNase for 2 h	-	[86]
Saphenous vein	0.25% trypsin with 0.02% EDTA for 5 min; 10 mmol/L Tris, 5 mmol/L EDTA for 72 h; frozen at −80 °C for 2 h and thawing at 37 °C for 30 min	50 mL 10 mM ethylenediaminetetraactic acid	[87]
Wistar rat kidney	Perfusated by 1% SDS	5 mL of papain solution for 24 h	[88]
Porcine kidney	Repetitive cycle of: perfused with 0.5 M NaCl solution for 30 min; 0.5% SDS solution for 30 min; deionized (DI) water for 30 min	Lyophilize and mechanically ground into fine powders	[89]
Rat kidney	Perfused with 1% SDS for 4 h or 1% SLES for 6 h	-	[90]
Rat kidney	perfused with 1% SDS for 3 h and 1% Triton X-100 for 16 h	-	[91]
Rabbit kidney	Perfused with 1% SDS for 90 h, 2% Triton X-100 for 12 h	-	[92]
Rat kidney	Perfused with 1% SDS for 48 h, 0.2 mg/mL deoxyribonuclease I and 10 mM MgCl_2_ for 16 h	-	[93]
Rhesus monkey kidney	Perfused with 1% SDS and 1% Triton X-100	-	[94]
Porcine kidney	Perfused with 1% SDS for 28 h, 1% Triton X-100 for 2 h	Incubation with papain extraction reagent for 3 h	[95]
Porcine skin	0.25% trypsin for 6 h; 0.1% SDS in 0.26% EDTA with 0.69% Tris for 6 h; 1% Triton X-100 in 0.26% EDTA with 0.69% Tris for 12 h	Lyophilized and dried for 72 h	[96]

## Data Availability

Not applicable.

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
