# Peer review of "Three-Dimensional Bioprinting of Decellularized Extracellular Matrix-Based Bioinks for Tissue Engineering"

_molecules, 2022, doi:10.3390/molecules27113442_

Round 1

Reviewer 1 Report

The authors thoroughly reviewed the recent advances in bioprinting of decellularized ECM-based bioinks for tissue engineering. The author's work is a timely topic and highly valuable. The authors addressed recent trends in natural ECM’s roles in cellular activities, decellularization treatment procedures and subsequent evaluations for quality control of dECM, and insights for future directions. Though the authors have presented an excellent framework for this review, I have the following points on content that may need to be addressed before further consideration:

I could easily find many published review articles on 3D bioprinting of dECM for tissue engineering. It is not clear what is different or unique about the article. As the authors described, the fabrication of ECM-based scaffold materials has been a significant impediment to clinics. I would suggest adding a section on these challenges in detail. These can include toxicity, immunities, degradations, mechanical properties, etc.

Also, what are the current challenges in the bioprinting of dECM bioinks, such as resolutions, printability, crosslinking, etc.?

As the authors pointed out, issues on the 3D printing of microvascular network structures need to be solved in the future direction. But, it is not clear how dECM bioprinting can be solved these challenges. Can it be improved by adding new functional nanomaterials or 3D printing methods? It might be ideal to add some detail and specific suggestions for future direction in at least a couple of paragraphs. 

Author Response

Dear Reviewer,

Re: Manuscript ID: molecules-1717821

The authors highly appreciate your valuable comments and suggestions during the peer-review process. We have carefully revised the manuscript according to your suggestions. Moreover, our replies and explanation point-to-point to each of the reviewer's comments are as given below and the changes were highlighted in the manuscript. Accordingly, we have now provided our revised versions of manuscript, i.e., Annotated version, in which all changes made are easily identifiable using the track changes function and a clean version for production along with this response letter.

Moreover, our manuscript have checked by Prof. Ranjith Kumar Kankala who is a native English speaker.

Best regards

Chaoping Fu

Reviewer 2 Report

The presented review entitled “Three-Dimensional Bioprinting of Decellularized Extracellular Matrix Based Bioinks for Tissue Engineering” authored by Zhang et al. represents a summary of the importance of decellularized extracellular matrix (ECM) for tissue engineering applications. In this context, the authors started with an introduction about the role of ECM and its composition, followed by different decellularization techniques used so far to prepare biological scaffolds providing different concrete examples. Additionally, the authors explained the different bioprinting techniques able to be applied to produce dECM for further regenerative applications. Finally, the authors provided examples from previously works of bioprined dECM bioinks for different tissues and organs. The review sounds interesting, designed and well written. The manuscript can be considered for publication after some minor revisions to be carried out from the authors:

  • Figure 2. The written terms within the Figure are not easily readable. The authors are invited to change the font of the text.
  • A lot of information does not have any reference. The authors are invited to add the references to support their statements. For instance, lines 141-143.
  • Table 1, the authors are invited to uniform the time as for example in some context it was written (1d) and in other context (24h). Please uniform.
  • Table 1 should be reformatted in which the authors can render it easier for the readers by following a specific order according to tissue types (heart, cartilage tissue, liver, etc.)
  • Considering the extrusion-based bioprinting as most used technique, the authors should emphasize this aspect by adding more information about the different conducted studies. An important feature to be discussed in this paragraph is related to the extrusion of dECMs. The dECM should be mixed with other synthetic materials to reach an adequate viscosity for a better extrusion? Please discuss.

Author Response

Dear Reviewer,

Re: Manuscript ID: molecules-1717821

The authors highly appreciate your valuable comments and suggestions during the peer-review process. We have carefully revised the manuscript according to your suggestions. Moreover, our replies and explanation point-to-point to each of the reviewer's comments are as given below and the changes were highlighted in the manuscript. Accordingly, we have now provided our revised versions of manuscript, i.e., Annotated version, in which all changes made are easily identifiable using the track changes function and a clean version for production along with this response letter.

Best regards,

Chaoping Fu

Round 2

Reviewer 1 Report

The revised manuscript can be accepted.